# Multiplex live single-cell transcriptional analysis demarcates cellular functional heterogeneity

**Ayhan Atmanli[1,2,3], Dongjian Hu[1,2,4], Frederik Ernst Deiman[1,2], Annebel Marjolein van de Vrugt[1,2], François Cherbonneau[1], Lauren Deems Black III[3,5], Ibrahim John Domian[1,2,6]\***

[1]Cardiovascular Research Center, Massachusetts General Hospital, Boston, United States; [2]Harvard Medical School, Boston, United States; [3]Department of Biomedical Engineering, Tufts University, Medford, United States; [4]Department of Biomedical Engineering, Boston University, Boston, United States; [5]Sackler School of Graduate Biomedical Sciences, Tufts University School of Medicine, Boston, United States; [6]Harvard Stem Cell Institute, Cambridge, United States

**Abstract** A fundamental goal in the biological sciences is to determine how individual cells with varied gene expression profiles and diverse functional characteristics contribute to development, physiology, and disease. Here, we report a novel strategy to assess gene expression and cell physiology in single living cells. Our approach utilizes fluorescently labeled mRNA-specific anti-sense RNA probes and dsRNA-binding protein to identify the expression of specific genes in real-time at single-cell resolution via FRET. We use this technology to identify distinct myocardial subpopulations expressing the structural proteins myosin heavy chain $\alpha$ and myosin light chain 2a in real-time during early differentiation of human pluripotent stem cells. We combine this live-cell gene expression analysis with detailed physiologic phenotyping to capture the functional evolution of these early myocardial subpopulations during lineage specification and diversification. This live-cell mRNA imaging approach will have wide ranging application wherever heterogeneity plays an important biological role.
DOI: https://doi.org/10.7554/eLife.49599.001

\*For correspondence:
domian@mgh.harvard.edu

## Introduction

A hallmark of development and disease is the cellular phenotypic diversification required for three-dimensional tissue structures. Cellular heterogeneity demonstrably contributes to the developmental dynamics of various types of stem cells (*Dulken et al., 2017*; *Kumar et al., 2014*; *Wilson et al., 2015*), neurons (*Sandoe and Eggan, 2013*) and cancer (*Meacham and Morrison, 2013*). In the heart, the coordinated differentiation, lineage diversification, and functional maturation of heterogeneous populations of cells is a prerequisite for the proper development of coordinated electrical and contractile function. Multiple cardiac myocyte lineages and sublineages, along with endothelial cells, smooth muscle cells and cardiac fibroblasts must interact in a cohesive program to form the mature four-chambered adult heart (*Bu et al., 2009*; *Domian et al., 2009*). Advances in pluripotent stem cell (PSC) biology open unprecedented avenues for the study of human cellular differentiation, physiology, and pathophysiology in vitro (*Lan et al., 2013*) and also underscore the heterogeneity of clinically important cell types (*Bryant et al., 1997*; *Burridge et al., 2014*; *Cordeiro et al., 2004*; *Lian et al., 2012*). This cellular heterogeneity along with an the inherent difficulty of examining real-time gene expression of single living cells poses a major limitation in the understanding of the complex biological processes that underlie development and disease.

Single-cell transcriptional profiling initially via multiplex qPCR analysis and more recently via whole transcriptome sequencing has provided insight into how intracellular signaling is regulated at the single-cell transcriptional level during cardiac development (*Cui et al., 2019*; *DeLaughter et al., 2016*; *Friedman et al., 2018*; *Li et al., 2016*; *Sahara et al., 2019*). Despite this progress, whole genome expression analysis does not allow for concurrent physiological assessment of single living cells and consequently, the functional significance of single-cell transcriptomic heterogeneity remains unclear. The live-cell identification of distinct cell populations has most commonly been accomplished with gene expression assays that rely on the detection of fluorescent reporter proteins under the transcriptional control of the gene of interest. Accordingly, these approaches require the generation of transgenic animals (*Domian et al., 2009*; *Wu et al., 2006*) or embryonic stem cell lines (*Elliott et al., 2011*; *Klug et al., 1996*) to isolate and study discrete subsets of cells with specific gene expression profiles. These methods are cumbersome, time consuming, and expensive and therefore allow for only a limited number of genes to be examined at a time. Technical advances have facilitated live-cell mRNA imaging by detecting gene transcripts via nucleic acid (*Santangelo et al., 2009*; *Tyagi and Kramer, 1996*; *Vargas et al., 2011*) or protein probes (*Bertrand et al., 1998*; *Nelles et al., 2016*; *Ozawa et al., 2007*). However, several drawbacks of these existing techniques such as genetic encoding of target mRNA and reporter protein, the necessity to target multiple binding sites, complexity of probe design and cellular delivery and low sensitivity (*Armitage, 2011*; *Tyagi, 2009*) have prevented their widespread use (*Table 1*).

Herein we describe a novel Förster Resonance Energy Transfer (FRET)-based technology for the *m*ultiplex *a*nalysis of *g*ene expression in *i*ndividual living *c*ells (termed MAGIC, patent application pending; *Atmanli and Domian, 2016*) (*Figure 1A*). Our technology enables the real-time assessment of transcript expression in single living cells. We functionally characterize distinct myocardial subpopulations derived from human PSC (hPSC) in real-time and describe marked differences in cell physiology among different cardiac sublineages. Our observations reinforce the importance of live-cell lineage assignment in a heterogeneous assembly of single living cells, and we anticipate that our technology could have wide-ranging application in any biological system.

## Results

### Strategy for the detection of transcripts in single living cells

Our approach uses two components to detect the expression of specific genes of interest (*Figure 1A*). First, we use fluorescently-labeled 20-mer antisense RNA probes (termed MAGIC Probes) designed to hybridize with specific intracellular RNA targets to result in fluorescent dsRNA complexes (*Table 2*). Second, we engineered a fluorescently-labeled dsRNA-binding protein (termed

**Table 1.** Comparison of MAGIC with other live-cell mRNA imaging technologies.

| | Advantages | Disadvantages |
|---|---|---|
| Nucleic Acid Probes | Most established approach | Complexity of probe design and cellular delivery |
| | Single-molecule sensitivity achievable | Need to screen many probes for specificity and sensitivity |
| | Cell isolation via FACS | Probe sequestration and false-positive signals |
| Protein Probes | Single-molecule sensitivity | Genetic encoding of target RNA and reporter protein |
| | Study of RNA dynamics | Multiple binding sites necessary |
| | | Low sensitivity |
| MAGIC | Imaging of transcription factors | Complexity of MAGIC Probe production |
| | Double detection with MAGIC Factor and Probes increases specificity | Efficient transfection of MAGIC Factor and Probes into the same cell required |
| | Coupling with cell physiology assays | |

DOI: https://doi.org/10.7554/eLife.49599.002

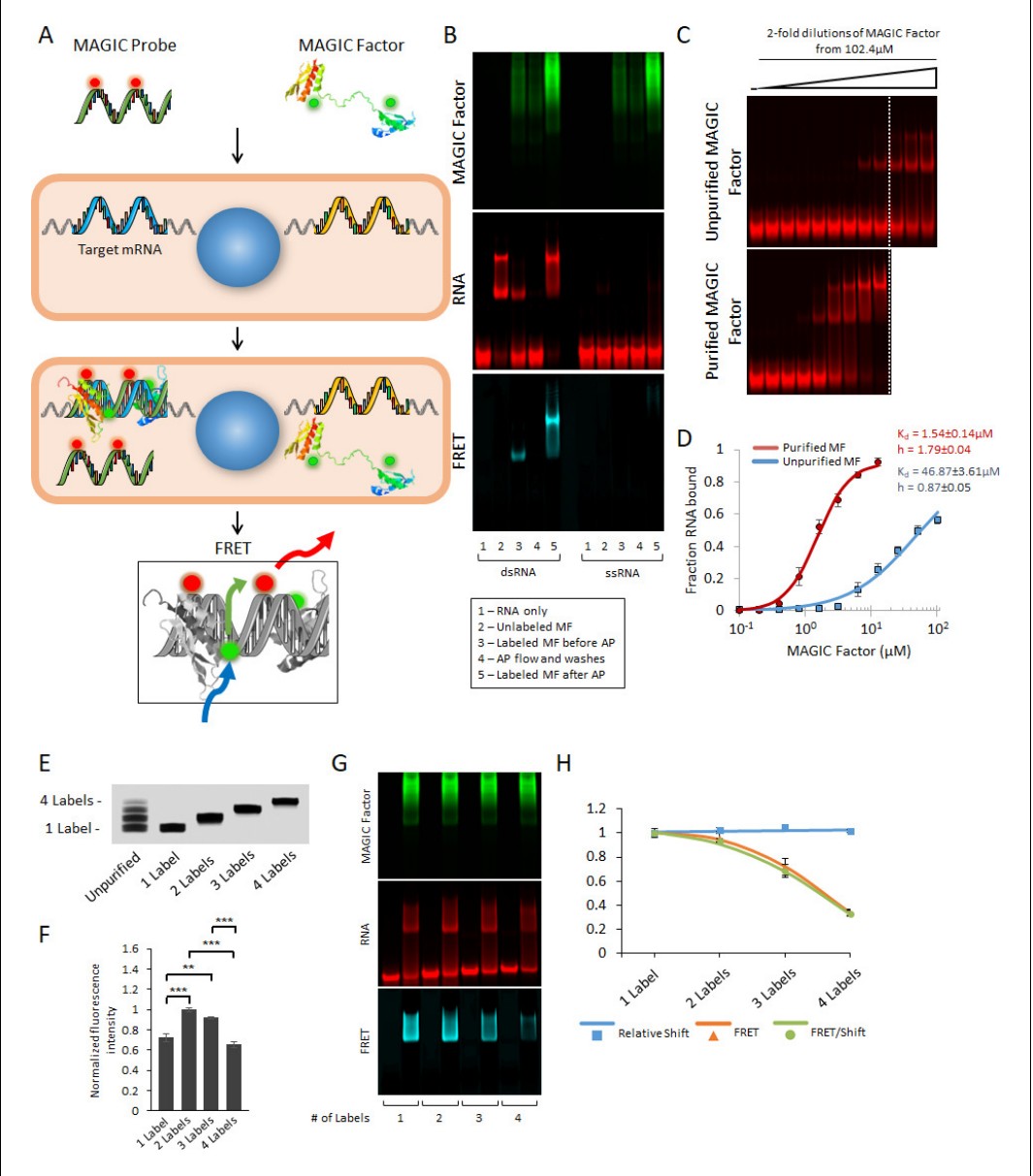

**Figure 1.** Strategy for the multiplex analysis of gene expression in individual living cells (MAGIC). (**A**) MAGIC anti-sense RNA Probes are generated by in vitro transcription using T7 phage polymerase and fluorescently-labeled using aminoallyl-modified UTP nucleotides. MAGIC Factor is the recombinant dsRNA-binding domain of human protein kinase R that has been fluorescently labeled and affinity purified against dsRNA. Both the probe and the protein are delivered into living cells by transfection. Upon cellular delivery, MAGIC Probes hybridize to their target gene and generate an RNA-RNA duplex. This enables MAGIC Factor to bind to the newly formed dsRNA, thereby bringing the donor fluorophore on the protein to come into close contact with the acceptor fluorophore on the RNA probe and for FRET to occur (blue arrow: excitation of the donor; green arrow: energy transfer; red arrow: emission of the acceptor). Cells expressing the gene of interest are identified by assaying the FRET signal. (**B**) Affinity purification of fluorescent MAGIC Factor restores its binding affinity. MAGIC Factor was fluorescently-labeled with Alexa Fluor 488 and then reacted with dsRNA-coupled agarose beads to separate binding, functional protein from non-binding, non-functional protein. An electrophoretic mobility shift assay (EMSA) of dsRNA and ssRNA labeled with Alexa Fluor 647 and reacted with MAGIC Factor is shown in the RNA, protein and FRET channels. (**C**) Unpurified and purified MAGIC Factor were reacted at increasing concentrations with a fixed concentration of dsRNA (200 nM) and run on a native 12% polyacrylamide gel. The dashed lines represent the cutoff after which increasing concentrations of only unpurified MAGIC Factor were reacted with dsRNA. (**D**) Quantification of the binding affinity (represented as dissociation constant Kd) and Hill coefficients of unpurified

*Figure 1 continued on next page*

*Figure 1 continued*

and affinity purified, fluorescent MAGIC Factor (MF) from the EMSA in (**C**). Affinity purification of MAGIC Factor resulted in a 30-fold increase in binding affinity to dsRNA. (**E**) In vitro transcribed and fluorescently-labeled 20-mer RNA probe was gel purified to obtain one-, two-, three- and four-labeled RNA probes. (**F**) The fluorescence intensities of equimolar concentrations of purified probes with one to four labels were measured using a spectrophotometer. (**G**) 20-mer RNA probes with one to four labels were reacted with unlabeled sense probes to generate dsRNA. Representative EMSA with MAGIC Factor is shown in the RNA, protein and FRET channels. Note that the appearance of MAGIC Factor as multiple bands is likely due to the use of an NHS-ester dye to attach Alexa Fluor 488 to the protein. Dependent on the exact location of the fluorophore molecule, each single protein molecule likely runs differently on the native polyacrylamide gel. (**H**) The relative shift of fluorescent dsRNA, the FRET intensity of shifted dsRNA and the FRET/shift ratio for one to four labeled RNA probes were quantified from the EMSA in (**G**). Quantified data are shown as mean ± s.e.m. \*\*p<0.01 and \*\*\*p<0.001.
DOI: https://doi.org/10.7554/eLife.49599.003
The following figure supplement is available for figure 1:

**Figure supplement 1.** Optimization of fluorescence labeling of MAGIC Factor and MAGIC Probes.
DOI: https://doi.org/10.7554/eLife.49599.004

MAGIC Factor) to specifically detect MAGIC Probes that are bound to their target. The MAGIC Factor was engineered by genetically and chemically altering the dsRNA-binding domain (dsRBD) of human protein kinase R (PKR). The dsRBD of PKR consists of two dsRNA-binding motifs encompassing amino acids 1–169 and binds to dsRNA with similar efficiency as the full-size parent protein (*Cosentino et al., 1995*; *Green and Mathews, 1992*). It exerts a specific interaction with dsRNA with no appreciable binding to dsDNA, ssDNA, ssRNA or RNA-DNA hybrids (*Bevilacqua and Cech, 1996*; *Nallagatla and Bevilacqua, 2008*; *Patel et al., 2012*). This interaction is sequence-independent and requires a minimum of 18–20 bp of dsRNA to achieve a robust interaction (*Bevilacqua and Cech, 1996*; *Ucci et al., 2007*). In order to detect the expression of specific RNA in single living cells, MAGIC Probes and MAGIC Factor are transiently co-delivered to individual living cells. In the presence of target RNA, MAGIC Probes hybridize to generate a fluorescent dsRNA complex. The subsequent binding of MAGIC Factor to the dsRNA complex brings both fluorophores into juxtaposition and can be detected by FRET imaging.

To obtain highly fluorescent and fully functional MAGIC Factor, we expressed the MAGIC Factor in a recombinant bacterial expression system and labeled it with Alexa Fluor 488. This fluorophore serves as a robust FRET donor due to its high quantum yield, photostability and low phototoxicity (*Dempsey et al., 2011*). Initial attempts at fluorescently labeling the MAGIC Factor through conjugation to its thiol or carboxylic acid groups were unsuccessful (data not shown) suggesting that these sites are not available for chemical modification. We therefore labeled MAGIC Factor at its primary

**Table 2.** Sequences of MAGIC probes and corresponding target sites.

| MAGIC Probe Target | Sequence | Target site |
|---|---|---|
| Human β-actin | 5'-GGATAGCACAGCCTGGATAG-3' | 507-488 |
| Human myosin heavy chain α (MHCα) | 5'-GGCACCAATGTCACGGCTCT-3' | 5864-5845 |
| Human myosin light chain 2a (MLC2a) | 5'-GGCCTGCTTGGTGGCTGCCA-3' | 66-47 |
| Human NKX2-5, Probe 1 | 5'-GGCTGCGCTGCTGCTGTTCC-3' | 308-289 |
| Human NKX2-5, Probe 2 | 5'-GGACGTGAGTTTCAGCACGC-3' | 766-747 |
| Human NKX2-5, Probe 3 | 5'-GCGTTATAACCGTAGGGATT-3' | 984-965 |
| Control Probe | 5'-GGATCGTGACTAGATCGTCA-3' | |
| Labeled ssRNA for EMSAs | 5'-GGATGAGTCACTGCCTAGCC-3' | |
| Unlabeled ssRNA complementary to probe above | 5'-GGCTAGGCAGTGACTCATCC-3' | |
| Labeled ssRNA for multiplex spectral FRET | 5'-GGAAGTAGCACAGTCCAGAC-3' | |
| Unlabeled ssRNA for affinity purification, strand 1 | 5'-GAGTCCTTCCACGATAGACC-3' | |
| Unlabeled ssRNA for affinity purification, strand 2 | 5'-GGTCTATCGTGGAAGGACTC-3' | |

DOI: https://doi.org/10.7554/eLife.49599.005

amino groups via N-hydroxysuccinimide (NHS) ester chemistry using molar ratios ranging from 0:1 and 6:1 between dye and protein. While this approach yielded a highly fluorescent protein without causing chemical degradation (*Figure 1—figure supplement 1A*), it resulted in a reduction of the protein's ability to bind fluorescent dsRNA (*Figure 1—figure supplement 1B*). As a consequence, the generation of a robust FRET signal was hindered (*Figure 1—figure supplement 1C*).

In order to purify functional from non-functional protein after fluorescent labeling, we affinity-purified the protein based on its ability to bind dsRNA. We linked synthetic dsRNA to agarose beads and reacted fluorescent MAGIC Factor with them. Using an increased concentration of potassium chloride, we retrieved the functional protein and then verified that the engineered protein specifically binds to dsRNA over ssRNA (*Figure 1B* and *Figure 1—figure supplement 1D*). Quantification of protein concentration in the individual fractions showed that only 4% of MAGIC Factor had been able to bind dsRNA after fluorescent labeling, but prior to further purification. As a result of using dsRNA-coupled beads, we were able to obtain fully functional, fluorescent MAGIC Factor. We then quantified the binding affinity of unpurified and affinity purified, labeled MAGIC Factor and found that affinity purification results in a ~ 30 fold increase in binding affinity, as shown using the corresponding $K_d$-values (*Figure 1C and D*). Thus, we have genetically and chemically engineered a novel fluorescentlylabeled dsRNA-binding protein that binds specifically to dsRNA but not ssRNA. Binding of this engineered protein to fluorescently labeled dsRNA results in a robust FRET signal.

Our next goal was to obtain highly fluorescent and fully functional MAGIC Probes. The probes consist of 20-mer RNA generated through standard in vitro transcription using T7 phage polymerase and using aminoallyl-modified uridine bases. Fluorescent labeling of the RNA oligonucleotide was then carried out by an NHS ester reaction with Alexa Fluor dyes with high extinction coefficients to provide optimum acceptor dyes for FRET (Alexa Fluor 546, 594 and 647). Since the chemical modification of RNA has the potential to alter fluorescence and hybridization kinetics of the probe (*Cox and Singer, 2004*), we evaluated the effects of the number of fluorescent dyes on each 20-mer RNA molecule. We fluorescently-labeled the RNA and purified one-, two-, three- and four-labeled RNA from a denaturing polyacrylamide gel (*Figure 1E*). We found a marked difference in the resulting fluorescence intensity between the differently labeled RNA probes, suggesting that over-labeling of MAGIC Probes results in fluorescence quenching (*Figure 1F*). When we hybridized the fluorescent RNA to its complementary RNA at 37°C, we found that only the four-labeled RNA exhibited significantly reduced hybridization kinetics *Figure 1—figure supplement 1E and F*). We then investigated whether the number of fluorescent dyes might affect the binding ability of MAGIC Factor and found no difference among the four RNA (*Figure 1G and H*). However, we found that the stronger labeled the RNA, the less the resulting FRET between MAGIC Factor and the RNA became, suggesting that the fluorescence quenching also affected the generation of FRET. Based on these results, we determined that one- and two-labeled MAGIC Probes are optimal for in vitro live-cell FRET imaging.

## In situ detection of transcripts in single living and fixed cells

In initial proof-of-principle experiments, we sought to visualize the β-actin mRNA in single living hPSC-derived cardiac myocytes (hPSC-CMs) using Alexa Fluor 647-labeled MAGIC Probes, similarly to previously reported live-cell mRNA detection approaches (*Santangelo et al., 2009*; *Tyagi and Alsmadi, 2004*). We first turned our attention to identifying the optimal conditions for the delivery of both MAGIC Factor and MAGIC Probes into single living cells. Since the efficiency of protein and RNA delivery is dependent on the construct and cell type of interest, we tested a range of commercially available transfection reagents for DNA, siRNA and proteins for their ability to efficiently deliver both constructs into single hPSC-CMs. We optimized our transfection protocol to achieve robust intracellular fluorescence signal intensities of MAGIC Factor and MAGIC Probes while maintaining cell viability above 95% (*Figure 2—figure supplement 1A–I*). Importantly, we were able to achieve high transfection efficiencies for both MAGIC Factor and Probes (>85% each), further minimizing the number of cells that are either not transfected or transfected with only one construct. Using our optimized transfection protocol, we delivered a novel β-actin MAGIC Probe along with MAGIC Factor into hPSC-CMs and found that our technology provides a robust FRET signal compared to a control probe that did not have an intracellular target (*Figure 2A*). We quantified the FRET signal using an established method to subtract spectral bleedthrough (corrected FRET, cFRET) (*Xia and Liu, 2001*) and found a robust increase in the calculated FRET signal when using the antisense probe (*Figure 2B*). To account for intercellular differences in cell size, we also quantified the

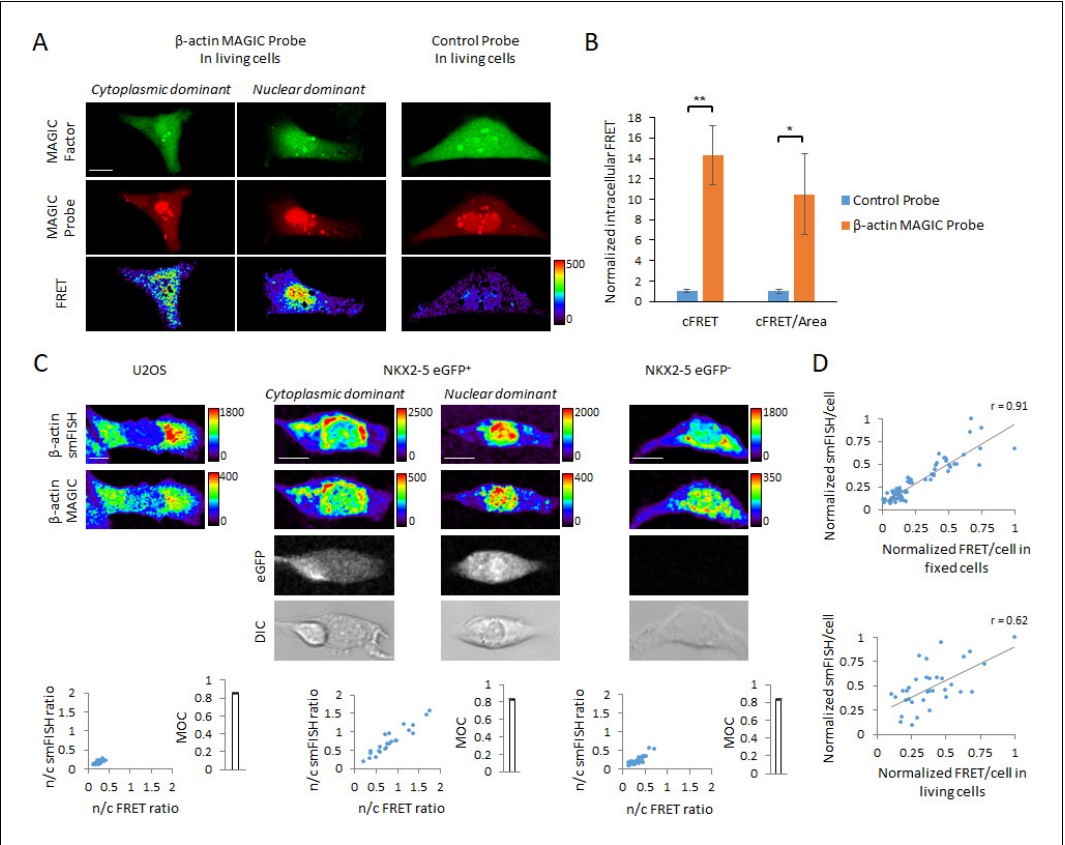

**Figure 2.** In situ visualization of transcripts in single living and fixed cells using MAGIC. (**A**) A novel MAGIC Probe against the human β-actin mRNA or control probe were delivered into single living hPSC-CMs together with MAGIC Factor. Both probes were labeled with Alexa Fluor 647 and MAGIC Factor was labeled with Alexa Fluor 488. The cells were assessed by confocal microscopy and the FRET images corrected for spectral bleed-through from both donor and acceptor. The resulting FRET image is shown in pseudocolor-coding. We found two different phenotypes of the β-actin mRNA localization: cells with cytoplasmic distribution and cells with nuclear localization of the mRNA. (**B**) The FRET intensity of assessed single cells was quantified by removing spectral bleed-through from both donor and acceptor (corrected FRET, cFRET) and further normalizing to cell area (cFRET/Area). $n \geq 22$ single cells per group. (**C**) U2OS cells, a pure population of cardiac myocytes (NKX2-5 eGFP[+]) and a non-cardiogenic cell population (NKX2-5 eGFP[-]) were fixed and subjected to simultaneous smFISH and MAGIC of the β-actin mRNA. The β-actin smFISH probes were labeled with Quasar 670, while MAGIC Factor was labeled with Alexa Fluor 488 and the β-actin MAGIC Probe was labeled with Alexa Fluor 546. From the smFISH and FRET images, the nuclear-to-cytoplasmic fluorescence intensity in each cell of each cell type was measured and the degree of co-localization quantified (Manders' Overlap Coefficient, MOC). The cells were assessed by confocal microscopy and the FRET images corrected for spectral bleed-through from both donor and acceptor. The resulting FRET image is shown in pseudocolor-coding. $n \geq 23$ single cells per group. (**D**) (Top) Normalized smFISH/cell versus normalized FRET/cell (from (**C**)) are plotted and show a robust positive correlation between both assays. Each point represents a single cell. (Bottom) MAGIC was performed on single living cells first and the same cells subjected to smFISH after fixation. Normalized smFISH/cell versus normalized FRET/cell are plotted and show that MAGIC can be used to infer to the level of mRNA expression in single living cells. Each point represents a single cell. Note that thresholded pixels are excluded from FRET images and thus appear as black signals. Further details on post-image processing are included in the Methods. Scale bars 25 μm. Quantified data are shown as mean ± s.e.m. *p<0.05 and **p<0.01.

DOI: https://doi.org/10.7554/eLife.49599.006

The following figure supplements are available for figure 2:

**Figure supplement 1.** Optimization of cellular delivery of MAGIC Factor and MAGIC Probes.
DOI: https://doi.org/10.7554/eLife.49599.007

**Figure supplement 2.** Cytoplasmic and nuclear localization of β-actin mRNA.
DOI: https://doi.org/10.7554/eLife.49599.008

*Figure 2 continued on next page*

*Figure 2 continued*

**Figure supplement 3.** Effect of cellular delivery of MAGIC Factor and MAGIC probes on cell viability, and mRNA and protein levels.

DOI: https://doi.org/10.7554/eLife.49599.009

cFRET signal normalized to cell area and confirmed the robust FRET signal when using the antisense probe.

Intriguingly, we found two different phenotypes of the β-actin mRNA localization in hPSC-CMs: cells with cytoplasmic distribution and cells with nuclear localization of the mRNA (*Figure 2A*). Of note, a number of studies have previously shown in a variety of cell lines that the β-actin mRNA is predominantly localized to the cytoplasm (*Ben-Ari et al., 2010*; *Buxbaum et al., 2014*; *Femino et al., 1998*; *Santangelo et al., 2009*). To corroborate our observation, we performed single-molecule fluorescence in situ hybridization (smFISH) of the β-actin mRNA simultaneously with MAGIC in fixed cells (*Figure 2C* and *Figure 2—figure supplement 2A*). Fixation resulted in significant variability in the fluorescence intensity and spectral bleed-through of Alexa Fluor dyes (*Figure 2—figure supplement 1J–K*). We therefore decided to react the MAGIC Factor and MAGIC Probe in single cells after fixation. We first tested our assay in U2OS cells, an established cell line in which the β-actin mRNA has been shown to be predominantly located in the cytoplasm (*Ben-Ari et al., 2010*; *Nelles et al., 2016*). The smFISH and the MAGIC FRET signals overlapped and confirmed the strong cytoplasmic distribution of the β-actin mRNA in U2OS cells with little nuclear localization, as shown by the nuclear-to-cytoplasmic (n/c) ratio. Of note, relative nuclear fluorescence signals were comparable between MAGIC and smFISH, demonstrating the specificity of our approach (*Figure 2—figure supplement 2B*). Because we performed our previous live-cell mRNA imaging assay in hPSC-CMs, we hypothesized that nuclear localization of the β-actin mRNA may be a specific phenotype in developing cardiac myocytes. To investigate this hypothesis, we differentiated reporter hPSCs toward the cardiac lineage, in which eGFP marks a pure cardiac myocyte population, whereas eGFP- cells are non-cardiogenic (*Elliott et al., 2011*). We first marked eGFP+ and eGFP- cells and then simultaneously probed each cell type with smFISH and MAGIC probes. We found that the eGFP+ cardiac myocytes had a high degree of variability with respect to the cytoplasmic distribution versus nuclear distribution of the β-actin mRNA. Strikingly, this finding was limited to the eGFP+ cardiac myocytes, as eGFP- cells predominantly trafficked the β-actin mRNA to the cytoplasm. Importantly, the MAGIC FRET signal co-localized with the smFISH signal in all cell types examined, as shown by the Manders' Overlap Coefficient (MOC). These results validate the specificity of MAGIC probes and the capacity of this technology to define the subcellular localization of specific transcripts.

We next examined if our live-cell mRNA imaging system can be used to quantitatively or qualitatively analyze the level of mRNA expression in single living cells. We first investigated whether the signal intensity in smFISH assays was linearly correlated with the MAGIC FRET signal intensity in fixed cells. As shown in *Figure 2D*, there was a strong correlation between the two fluorescent assays. We then performed MAGIC on β-actin in living cells first, then fixed and performed smFISH on the same cells. Again we found a positive correlation between the two assays, suggesting that the intensity of FRET signals in MAGIC probes can be used to infer relative gene expression levels. An interesting observation was that correlating MAGIC in living cells with smFISH in fixed cells resulted in a lower Pearson's correlation coefficient as compared with performing MAGIC and smFISH simultaneously in fixed cells. This may be due to higher autofluorescence levels in living cells, making the FRET/cell values in living cells less predictable than in fixed cells. Finally, we showed that co-transfection of MAGIC Probes and MAGIC Factor did not alter cell viability, target mRNA levels, or target protein levels (*Figure 2—figure supplement 3*), consistent with previous reports that unmodified RNAs result in no significant RNA interference and translation inhibition by steric hindrance (*Kole et al., 2012*).

Taken together, these results demonstrate that MAGIC technology enables the robust localized detection of specific mRNA transcripts in single living and fixed cells, that it enables the inference to gene expression levels, all without disrupting cell viability and gene expression levels.

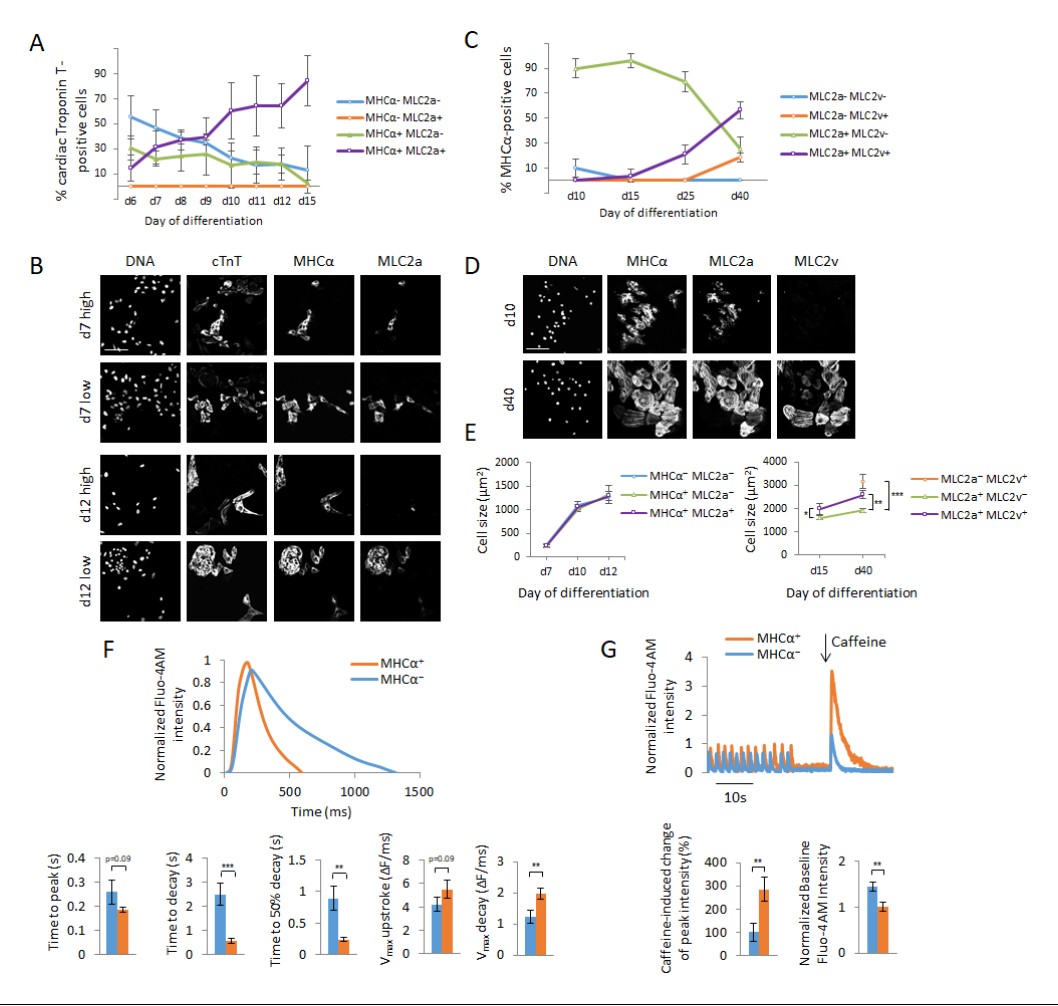

**Figure 3.** Cellular heterogeneity and functional assessment of hPSC-CMs using MAGIC. (**A**) hPSC-CMs were quantified for their expression of myosin light chain 2a (MLC2a) and myosin heavy chain alpha (MHCα) between days 6 and 15 of hPSC differentiation by immunofluorescence staining. Cardiac myocytes were identified by their co-expression of cardiac troponin T (cTnT). Quantitative assessment of protein expression is shown for days 6–15 of hPSC differentiation. (**B**) The replicates in (**A**) displayed a high degree of batch-to-batch variability in the co-expression of cTnT, MHCα and MLC2a. Representative images of cells from two different batches of each day 7 and day 12 of hPSC differentiation with a low and high degree of cellular heterogeneity are shown. A low degree of heterogeneity results in more uniform cell populations with most cardiac myocytes expressing MHCα and MLC2a, whereas a high degree of heterogeneity results in more diverse cell populations with most cardiac myocytes differentially expressing MHCα and MLC2a. (**C**) hPSC-CMs were quantified for their expression of MLC2a and myosin light chain 2 v (MLC2v) between days 10 and 40 of hPSC differentiation by immunofluorescence staining. Cardiac myocytes were identified by their co-expression of MHCα. Quantitative assessment of protein expression is shown for days 10–40 of hPSC differentiation. (**D**) Representative images of days 10 and 40 of hPSC differentiation of (**C**). (**E**) Cell size differences of myocardial sublineages of (**A**) and (**C**) at various time points of cellular differentiation were quantified (n = 227 (left panel) and n = 190 (right panel)). (**F**) Assessment of calcium handling of hPSC-CMs differing in their expression of MHCα. Single living hPSC-CMs were analyzed for MHCα gene expression using MAGIC first (same fluorophore combination as in (**F**)) and then loaded with the calcium indicator Fluo-4 AM to analyze their spontaneous $Ca^{2+}$ handling properties. Representative $Ca^{2+}$ transients and quantitative $Ca^{2+}$ kinetics of MHCα+ and MHCα− CMs are shown (n = 28 cells). (**G**) Assessment of calcium handling of hPSC-CMs differing in their expression of MHCα in response to caffeine. Single living hPSC-CMs were first analyzed for MHCα gene expression (same fluorophore combination as in (**F**)) and then loaded with the calcium indicator Fluo-4 AM. 10 mM of caffeine was added in calcium-free buffer during data acquisition. Representative $Ca^{2+}$ transients and quantitative $Ca^{2+}$ kinetics of MHCα+ and MHCα− CMs in response to caffeine are shown (n = 19 cells). Scale bars 100 μm. Quantified data are shown as mean ± s.e.m. *p<0.05, **p<0.01 and ***p<0.001.

*Figure 3 continued on next page*

*Figure 3 continued*

DOI: https://doi.org/10.7554/eLife.49599.010

The following figure supplement is available for figure 3:

**Figure supplement 1.** Development of MAGIC Probes against MHCα and MLC2a, and effect of MAGIC Probe transfection on calcium handling of hPSC-CMs.

DOI: https://doi.org/10.7554/eLife.49599.011

## Transcriptional and morphological characterization of cardiac sublineages

We next applied our live-cell mRNA imaging system to the functional analysis of different subsets of hPSC-CMs. Prior work from a number of laboratories demonstrated that transcript levels of myosin light chain 2a (MLC2a) and myosin heavy chain α (MHCα) increase during early cardiac differentiation of hPSC (*Burridge et al., 2014*; *Lian et al., 2012*), suggesting a key role for both structural proteins for proper development of cardiac contractile function. However, it is not known how the expression of these proteins interplays with cell physiology during early hPSC differentiation. To address this gap in knowledge, we first characterized the expression of these proteins at the single-cell level after the onset of beating between days 6 and 15 of hPSC differentiation toward the cardiac lineage. We found that MHCα and MLC2a are expressed sequentially, resulting in MHCα$^+$/MLC2a$^+$ and MHCα$^+$/MLC2a$^-$ hPSC-CMs (*Figure 3A and B*). Intriguingly, we found that while by day 15 the vast majority of hPSC-CMs were co-expressing MHCα$^+$ and MLC2a$^+$, no cardiac myocyte expressed MLC2a without also co-expressing MHCα at any time. Of important note, we also found a significant degree of batch-to-batch variability in the co-expression of these proteins (*Figure 3B*), further underscoring the importance of live-cell lineage assignment in real-time.

To investigate whether the expression of these proteins may confer chamber specificity, we characterized the expression of MHCα and MLC2a together with myosin light chain 2 v (MLC2v), a specific ventricular marker in the human heart (*Chuva de Sousa Lopes et al., 2006*), during early (up to day 15) and late (up to day 40) differentiation of hPSC. Of note, MLC2a is expressed in all chambers of the developing human heart (*Chuva de Sousa Lopes et al., 2006*; *van den Berg et al., 2015*). We found that MLC2v expression commences at day 15 of differentiation and significantly increases as differentiation continues (*Figure 3C and D*). Concomitantly, the expression of MLC2a starts to decrease beginning at day 15, leaving a mixed population of cells consisting of MLC2a$^+$/MLC2v$^+$, MLC2a$^+$/MLC2v$^-$ and MLC2a$^-$/MLC2v$^+$ hPSC-CMs at day 40 of differentiation. Of important note, these two markers were exclusively expressed in MHCα$^+$ cells and there were no hPSC-CMs that did not express either MLC2a or MLC2v at any time. Our results demonstrate that during early differentiation cardiac myocytes are yet unspecified with respect to cardiac chambers, as shown by the lack of MLC2v expression until day 15 of differentiation. Further, our data suggest that the expression of MHCα and MLC2a is not sufficient to deduce chamber specificity, in line with previous transcriptomic analysis of the developing human heart (*van den Berg et al., 2015*). We summarize the expected expression levels of MHCα, MLC2a and MLC2v in atrial versus ventricular cardiac myocytes during early and late differentiation based on our and previously described results (*Kamakura et al., 2013*; *van den Berg et al., 2015*) in *Table 3*.

We also analyzed the morphological differences of the various cardiac sublineages. Intriguingly, we found that during early cardiac differentiation up to day 15 cells do not increase in their size as they differentially express MHCα and MLC2a (*Figure 3E*). In contrast later in differentiation, the expression of MLC2v is accompanied with an increase in cell size consistent with prior reports that ventricular cardiac myocytes are larger than atrial cardiac myocytes (*Campbell et al., 1987*). Taken together, these dynamics during cardiac development demonstrate the necessity for live-cell lineage assignment to understand the cell physiology of individual types of cardiac myocytes. We therefore asked whether there are functional differences associated with MHCα and MLC2a expression during early cardiac cellular differentiation.

## Functional characterization of cardiac sublineages

To address this question, we targeted the MHCα (*Figure 3—figure supplement 1A and B*) and MLC2a (*Figure 3—figure supplement 1C and D*) mRNA individually in a pure hPSC-CM population

**Table 3.** Expected expression levels of MHCα, MLC2a and MLC2v in atrial and ventricular hPSC-CMs during early vs. late differentiation (*Kamakura et al., 2013*; *van den Berg et al., 2015*).

| | atrial | | ventricular | |
|---|---|---|---|---|
| | early | late | early | late |
| MHCα | + | +++ | + | +++ |
| MLC2a | + | +++ | + | ++ |
| MLC2v | - | - | - | +++ |

DOI: https://doi.org/10.7554/eLife.49599.012

The following source data is available for Table 3:

Source data 1. Summary of quantified data.
DOI: https://doi.org/10.7554/eLife.49599.013

Source data 2. Fluorescence intensity quantification of quantified confocal microscopy data for each channel (MAGIC Factor, MAGIC Probe, FRET).
DOI: https://doi.org/10.7554/eLife.49599.014

and demonstrated a high degree of specificity and sensitivity of MAGIC Probes after post-staining for MHCα and MLC2a protein, respectively. Using specific custom smFISH probes to quantify transcript levels of β-actin, MHCα and MLC2a in single hPSC-CMs we showed that MHCα and MLC2a mRNA counts are lower than β-actin mRNA counts, further demonstrating that our technology can detect lower abundant transcripts (*Figure 4—figure supplement 1*). The mRNA count of β-actin in hPSC-CMs was consistent with previous reports (*Femino et al., 1998*; *Santangelo et al., 2009*).

Since MHC proteins have previously been shown to play a critical role for the calcium handling of hPSC-CMs (*Lan et al., 2013*), we first investigated how MHCα expression correlates with cardiac myocyte function. We probed for MHCα$^+$ and MHCα$^-$ hPSC-CMs using MAGIC and then assessed the calcium handling of these cells using the calcium indicator Fluo 4-AM. Using the live-cell mRNA imaging system, we were able to show that MHCα$^+$ hPSC-CMs possess faster spontaneous Ca$^{2+}$ kinetics and in particular the ability to shuttle free calcium faster between contraction cycles (*Figure 3F*). The intracellular delivery of MAGIC Probes and MAGIC Factor did not affect cellular Ca$^{2+}$ dynamics (*Figure 3—figure supplement 1E*).

We then tested the effect of caffeine, a sensitizer of ryanodine receptors on the sarcoplasmic reticulum, on these cardiac subpopulations and found that hPSC-CMs expressing MHCα exhibited a significantly stronger calcium influx into the cytoplasm in response to caffeine than MHCα$^-$ hPSC-CMs (*Figure 3G*). These results show that MHCα$^+$ cardiac myocytes possess relatively more mature contractile apparatus than MHCα$^-$ cardiac myocytes, consistent with previous reports that older, more mature cardiac myocytes exhibit faster Ca$^{2+}$ kinetics (*Satin et al., 2008*).

Our findings show that MHCα$^+$ cardiac myocytes possess developmentally more advanced contractile apparatus than MHCα$^-$ cardiac myocytes, demonstrating how during early cardiac differentiation the step-wise expression of contractile proteins promotes cellular differentiation and function. Our findings are consistent with previous reports showing that early cardiac cellular differentiation is accompanied by increased expression of MHCα, further corroborating the validity of our approach (*Bizy et al., 2013*; *Piccini et al., 2015*). Taken together, we showed that the expression of a key player of a cellular differentiation program is directly linked with the functional outcome at single-cell resolution. These results demonstrate how progression in cellular differentiation is tightly associated with functional evolution.

## Detection of dynamic range of transcript levels

We next asked whether our technology can detect a dynamic range of transcript levels. To address this question, we subjected undifferentiated hPSCs, differentiating hPSCs at day 5 of cardiac differentiation as well as differentiated cardiac myocytes at days 8 and 20 of differentiation and subjected them to smFISH staining using our custom MHCα smFISH probes. By quantifying the smFISH fluorescence signals per single cell, we found that cells progressively increased the expression of MHCα during differentiation (*Figure 4A*). We then delivered MAGIC Factor and our specific MHCα MAGIC

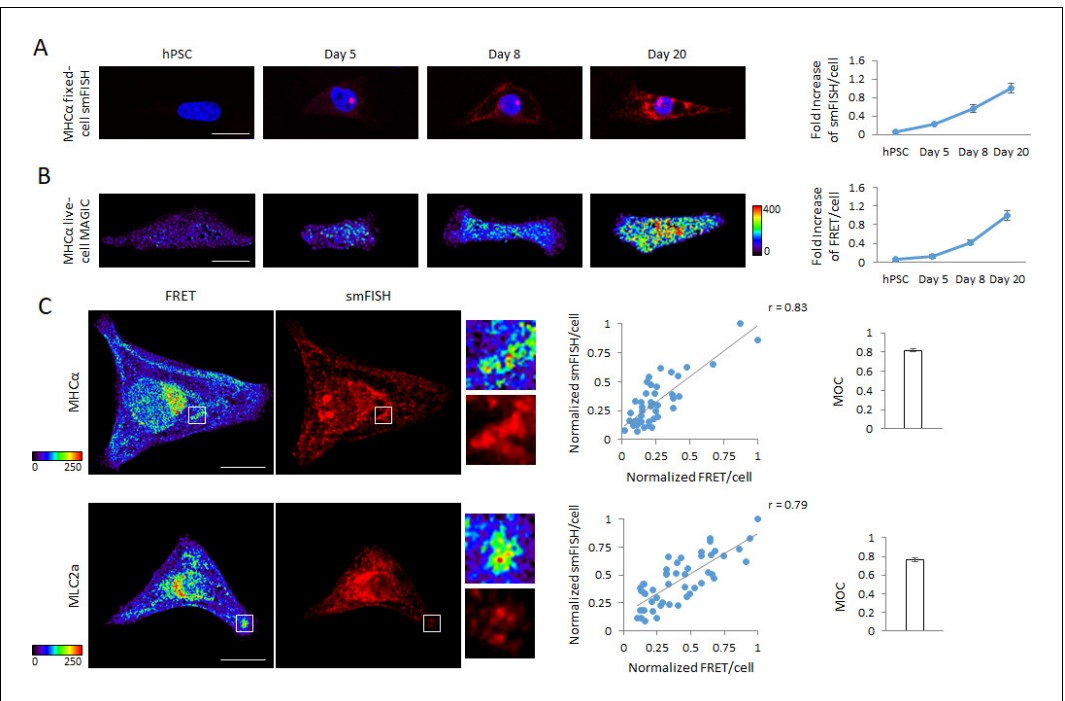

**Figure 4.** Detection of dynamic range of transcript levels using MAGIC. (**A**) Undifferentiated hPSCs, differentiating hPSCs at day 5 of cardiac differentiation as well as differentiated cardiac myocytes at days 8 and 20 of differentiation were fixed and stained with smFISH probes against the MHCα mRNA. Normalized smFISH/cell is plotted and shows an increase of transcript levels as cellular differentiation of hPSCs proceeds (n = 164 cells). (**B**) Living undifferentiated hPSCs, differentiating hPSCs at day 5 of cardiac differentiation as well as differentiated cardiac myocytes at days 8 and 20 of differentiation were transfected with MAGIC Factor and MAGIC Probe against the MHCα mRNA. Single living cells were assessed for FRET by confocal microscopy and the FRET images were corrected for spectral bleed-through from both donor and acceptor. The resulting FRET image is shown in pseudocolor-coding. Normalized FRET/cell is plotted and shows an increase of transcript levels as cellular differentiation of hPSCs proceeds, demonstrating that MAGIC is able to detect a dynamic range of transcript levels (n = 65 cells). (**C**) hPSC-CMs were subjected to simultaneous smFISH and MAGIC of the MHCα (top panel) and MLC2a (bottom panel) mRNA. smFISH/cell and FRET/cell are plotted and show a robust positive correlation between both assays. Each point represents a single cell. From the smFISH and FRET images, the degree of co-localization was quantified (Manders' Overlap Coefficient, MOC). The cells were assessed by confocal microscopy and the FRET images corrected for spectral bleed-through from both donor and acceptor. The resulting FRET image is shown in pseudocolor-coding. n ≥ 45 single cells per group. Note that thresholded pixels are excluded from FRET images and thus appear as black signals. Further details on post-image processing are included in the Materials and methods. Scale bars 25 µm (**A**,**B**) and 20 µm (**C**). Quantified data are shown as mean ± s.e.m.
DOI: https://doi.org/10.7554/eLife.49599.015
The following figure supplement is available for figure 4:

**Figure supplement 1.** smFISH imaging of β-actin, MHCα and MLC2a mRNA in hPSC-CMs and U2OS cells.
DOI: https://doi.org/10.7554/eLife.49599.016

Probe into single living undifferentiated hPSCs, differentiating hPSCs at day 5 of cardiac differentiation as well as differentiated cardiac myocytes at days 8 and 20 of differentiation and quantified the FRET per cell signals in single living cells (*Figure 4B*). Strikingly, using our technology, we were able to detect the same transcriptional dynamics of differentiating cells as using smFISH, demonstrating that our approach can be utilized to detect mRNA at the onset of transcription when they are present at less abundant levels. Further, we show that our technology enables the study of individual cells at different stages of cellular differentiation.

These findings were further corroborated by simultaneous detection of the MHCα or MLC2a mRNA by MAGIC and smFISH in fixed hPSC-CMs. Quantification of the FRET per cell and smFISH per cell intensities revealed not only a strong correlation between these two parameters but also a

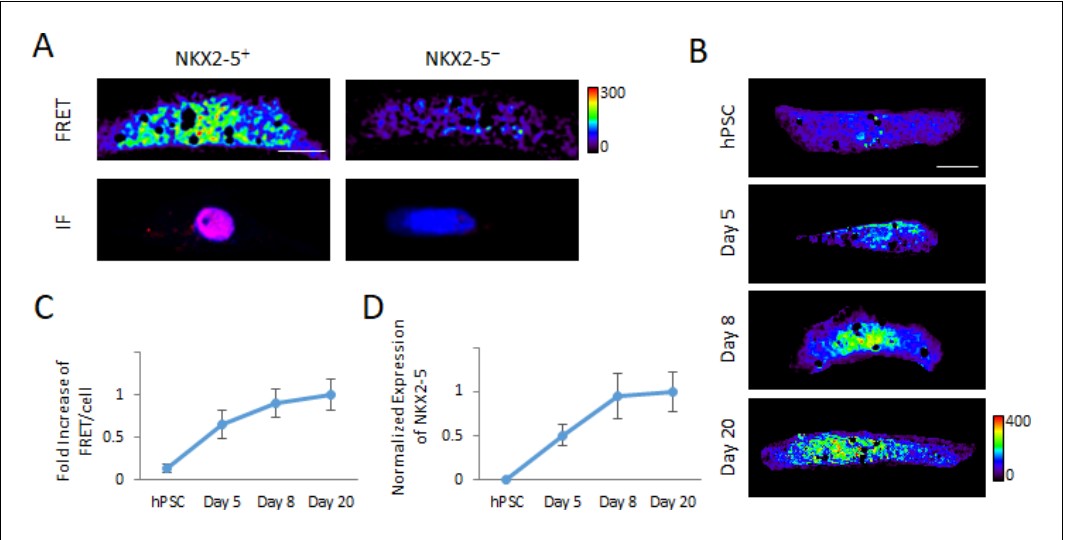

**Figure 5.** Detection of transcription factor mRNA using MAGIC. (**A**) Single living hPSC-CMs were first analyzed for NKX2-5 gene expression using three unique Alexa Fluor 594-labeled MAGIC Probes and Alexa Fluor 488-labeled MAGIC Factor, and then analyzed for NKX2-5 protein expression by immunofluorescence staining (IF). FRET images were corrected for spectral bleed-through from both donor and acceptor and the resulting FRET image is shown in pseudocolor-coding. (**B**) Living undifferentiated hPSCs, differentiating hPSCs at day 5 of cardiac differentiation as well as differentiated cardiac myocytes at days 8 and 20 of differentiation were transfected with MAGIC Factor and MAGIC Probes against the NKX2-5 mRNA. Single living cells were assessed for FRET by confocal microscopy and the FRET images were corrected for spectral bleed-through from both donor and acceptor. The resulting FRET image is shown in pseudocolor-coding. (**C**) Normalized FRET/cell from the cells in (**B**) is plotted and shows an increase of transcript levels as cellular differentiation of hPSCs proceeds, demonstrating that MAGIC is able to detect a dynamic range of transcript levels of low-abundant transcription factors (n = 80 cells). (**D**) Transcript levels of the NKX2-5 gene in undifferentiated hPSCs, differentiating hPSCs at day 5 of cardiac differentiation as well as differentiated cardiac myocytes at days 8 and 20 of differentiation were assessed via quantitative PCR and normalized against the housekeeping gene GAPDH (n = 3 independent experiments). Note that thresholded pixels are excluded from FRET images and thus appear as black signals. Further details on post-image processing are included in the Methods. Scale bars 15 μm. Quantified data are shown as mean ± s.e.m.
DOI: https://doi.org/10.7554/eLife.49599.017

wide range of transcriptional activity of the MHCα and MLC2a genes in hPSC-CMs that MAGIC was able detect, demonstrating that our technology is suited for the single-cell analysis of heterogeneous cell populations. Further, we were again able to show that FRET signals co-localize with smFISH signals, as shown by a highly localized FRET and smFISH fluorescence pattern as well as a computational pixel-by-pixel analysis (MOC) (*Figure 4C*).

## Detection of mRNA of transcription factors in single living cells

We next sought to visualize low-copy mRNA in single living cells. Because the transcription factor NKX2-5 plays a central role during cardiac cellular differentiation (*Wu et al., 2006*), we made unique MAGIC Probes against the NKX2-5 mRNA. We first tested whether our technology enables the detection of lower abundant transcripts, such as the ones from transcription factors, and found that when using a set of three different probes the NKX2-5 mRNA can be visualized in single living cells using MAGIC (*Figure 5A*). We then asked if MAGIC can be used to analyze the longitudinal expression of developmental transcription factors, such as NKX2-5, from undifferentiated hPSC to differentiated cardiac myocytes. We delivered MAGIC Factor and our specific NKX2-5 MAGIC Probes into single living undifferentiated hPSCs, differentiating hPSCs at day 5 of cardiac differentiation as well as differentiated cardiac myocytes at days 8 and 20 of differentiation and quantified the FRET per cell signals in single living cells (*Figure 5B and C*). Strikingly, using MAGIC we were able to replicate

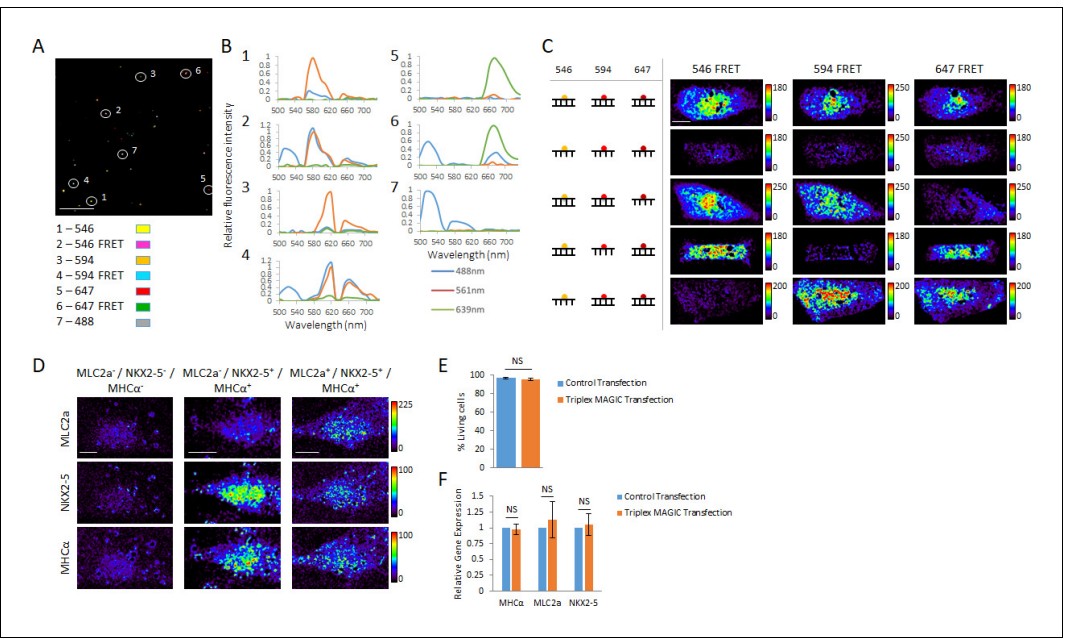

**Figure 6.** Multiplex spectral FRET imaging in single living cells. (**A**) Individual lipocomplexes containing only RNA labeled with Alexa Fluor 488, 546, 594, 647, 488 and 546, 488 and 594 or 488 and 647 were visualized by spectral imaging, resolving four individually labeled RNA constructs and three RNA FRET constructs. Image shown is a pseudo-colored merge image of all seven RNA constructs after linear unmixing data processing. Circles highlight individual lipocomplexes containing one of each RNA construct. (**B**) The spectral profiles of the highlighted, individual lipocomplexes in (**A**) were plotted from spectral imaging data, validating multiplexing potential of spectral FRET imaging. Each construct was excited with 488 nm, 561 nm and 639 nm lasers and their spectral profile between 500 and 740 nm recorded. (**C**) 20-mer dsRNA or ssRNA probes with no target in human cells and fluorescently labeled with Alexa Fluor 546, 594 or 647 were co-delivered into living hPSC-CMs together with MAGIC Factor as indicated. Spectral imaging and linear unmixing were performed on single living cells. (**D**) Single living hPSC-CMs were analyzed for MLC2a, NKX2-5 and MHCα expression using Alexa Fluor 546-, Alexa Fluor 594- and Alexa Fluor 647-labeled probes, respectively. Spectral imaging and linear unmixing were performed to identify individual cell subpopulations. The resulting individual FRET images are displayed in pseudo-coloring. (**E**) hPSC-CMs were control transfected or co-delivered with MAGIC Factor and MAGIC Probes against the MHCα, MLC2a and NKX2-5 mRNA. The viability of living cells in each group was assessed after transfection using a two-color fluorescence assay (n = 3 independent experiments). (**F**) hPSC-CMs were control transfected or co-delivered with MAGIC Factor and MAGIC Probes against the MHCα, MLC2a and NKX2-5 mRNA. Transcript levels of the three genes were assessed after transfection via quantitative PCR and normalized against the housekeeping gene GAPDH (n = 3 independent experiments). Note that thresholded pixels are excluded from FRET images and thus appear as black signals. Further details on post-image processing are included in the Methods. Scale bars 4 μm (**A**) and 15 μm (**C,D**). Quantified data are shown as mean ± s.e.m. NS, not significant.

DOI: https://doi.org/10.7554/eLife.49599.018

The following figure supplements are available for figure 6:

**Figure supplement 1.** Spectral imaging of complexed fluorescent RNA probes.
DOI: https://doi.org/10.7554/eLife.49599.019

**Figure supplement 2.** Spectral FRET imaging.
DOI: https://doi.org/10.7554/eLife.49599.020

the transcriptional dynamics of the longitudinal expression of NKX2-5 in differentiating hPSCs as compared with conventional quantitative PCR (*Figure 5D*). Taken together, our data show that MAGIC not only enables the detection of low-copy mRNA, such as the ones from transcription factors, but also to analyze the transcriptional dynamics of transcription factors during developmental processes.

# Multiplex spectral FRET imaging and assessment of gene expression in single living cells

Prior technologies aimed at targeting specific transcripts in living cells have mainly focused on a simplex approach (*Ban et al., 2013*; *Ozawa et al., 2007*; *Santangelo et al., 2009*). A major goal of this study was the simultaneous imaging of multiple transcripts in single living cells. We have performed FRET imaging in the previous results using the sensitized emission approach, in which the emission of the acceptor fluorophore using the donor excitation is visualized in a separate channel upon FRET. We reasoned that the spectral bleed-through from multiple fluorophores would limit the capability of this imaging technique to resolve a multiplex FRET approach. For assays employing more than one FRET pair we therefore used spectral imaging to enable a robust resolution of multiple FRET pairs in single living cells. In spectral imaging, the entire fluorescence spectrum is acquired upon laser excitation to form a so called 'lambda stack' consisting of multiple detection windows spanning typically 10–50 nm (*Zimmermann et al., 2003*). Using the spectral fingerprints of each fluorophore from corresponding single-labeled controls, the individual fluorophores in the experimental sample can be resolved through a process termed 'linear unmixing'. This process is based on the assumption that each fluorophore contributes in a linear manner to the fluorescence signal recorded in any given spectral detection channel. The relative contribution of each fluorophore is then used to unmix the composite fluorescence signal into individual images of each employed fluorophore. Because all light is collected for determining which fluorophore contributes to every given fluorescence signal, even fluorophores with minute differences in their emission spectra, such as GFP and FITC whose peak emission fluorescence are separated by only 7 nm can be resolved (*Haraguchi et al., 2002*). Spectral imaging has been highly validated in both conventional fluorescence and FRET imaging (*Sun et al., 2010*; *Woehler, 2013*; *Zimmermann et al., 2003*).

We tested whether we can faithfully image the three FRET pairs Alexa Fluor 488–546, 488–594 and 488–647. Because Alexa Fluor 488 has a quantum yield of 0.92 (*Dempsey et al., 2011*), we chose to use it as a common donor for all three acceptors used in this study. We prepared 20-mer RNA probes with two labeling sites and generated a total of seven probes including all three FRET probes and four single-labeled control probes. We recovered single- and double-labeled RNA from polyacrylamide gels. To image them on a laser scanning confocal microscope, we packed the probes in lipocomplexes. We found that individual complexation and sequential loading of the probes onto a plate prevented any co-localization of the lipocomplexes, ensuring that each fluorescence signal originates from a specific probe (*Figure 6—figure supplement 1A*). We loaded all seven probes onto a plate and acquired spectral images ranging from 500 to 740 nm (*Figure 6—figure supplement 1B*), and applied linear unmixing to resolve all seven distinct fluorescence signals (*Figure 6A* and *Figure 6—figure supplement 1C*). Tracking the spectral profile of distinct lipocomplexes confirmed their unique identity (*Figure 6B*). Because the resulting FRET image after linear unmixing still contains the acceptor bleed-through, we further removed it computationally. Importantly, we found using single-labeled control probes that the acceptor bleed-through remained stable with all three acceptors over a large range of fluorescence intensity (*Figure 6—figure supplement 2A–D*), making the computational removal of spectral-bleedthrough after linear unmixing a reliable method.

We next tested the suitability of spectral imaging to resolve fluorophores with close spectra in cellular assays and found that it provides a robust method to differentiate between Alexa Fluor 546 and 594, as shown by immunofluorescence staining of the nuclearly located NKX2-5 and the cytoplasmically located MHCα proteins with these fluorophores (*Figure 6—figure supplement 2E*). We then investigated whether spectral imaging allows to resolve the same three FRET pairs as above in single living cells using MAGIC. We prepared RNA probes labeled with Alexa Fluor 546, 594 or 647 and hybridized them with unlabeled, complementary RNA to form the corresponding dsRNA probes. We delivered different combinations of fluorescent dsRNA and ssRNA probes together with MAGIC Factor into hPSC-CMs and performed spectral imaging and linear unmixing on single living cells (*Figure 6C* and *Figure 6—figure supplement 2F–G*). Importantly, we found that multiplex RNA co-transfection approach yielded a 97% efficiency of the delivery of all three probes into individual cells (*Figure 6—figure supplement 2H*). Furthermore using spectral imaging, we were able to faithfully resolve the FRET signals from other fluorescence signals in single living cells.

Finally to demonstrate the multiplex potential of our technology, we delivered the MLC2a, NKX2-5 and MHCα MAGIC Probes into single living hPSC-CMs and identified distinct cardiac lineages

based on the expression of these three genes (*Figure 6D*). We further showed that the co-delivery of MAGIC Factor and three distinct MAGIC probes does not alter cell viability (*Figure 6E*) or mRNA levels (*Figure 6F*). Taken together, MAGIC coupled with spectral imaging provides a powerful tool for the multiplex imaging of at least three genes pairs in single living cells and for the study of the processes of lineage commitment and diversification in single living cells.

## Discussion

In this work, we presented a novel strategy to correlate gene expression and cell physiology in a heterogeneous assembly of single living cells. Previous studies aimed at targeting specific mRNA in living cells via nucleic acid or protein probes (*Bao et al., 2009*; *Bertrand et al., 1998*; *Nelles et al., 2016*; *Ozawa et al., 2007*; *Santangelo et al., 2009*). Our strategy utilizes a hybrid approach comprising fluorescent RNA probes and dsRNA-binding protein to enable FRET to discern hybridized probes within single living cells. We were able to assess the cell physiology of distinct myocardial sublineages during early cardiac differentiation from hPSC. To delineate the process of functional maturation in human cardiac myocytes, we studied how the transcriptional activation of a key contractility protein, MHCα, impacts the calcium handling capacity of hPSC-CMs. This approach has the broader potential to study how the expression of specific genes regulates single-cell physiology. As more single-cell transcriptional profiling and related technologies identify new candidate lineage-specific markers, it is increasingly important to validate the functional relevance of these markers by single-cell functional assays, such as the one presented herein.

Cellular heterogeneity has been a persistent challenge in biomedical research. Pronounced cell-to-cell variation with respect to morphology and function is evident within multiple tissues and cell types, such as stem cells (*Dulken et al., 2017*; *Kumar et al., 2014*; *Wilson et al., 2015*), cardiac myocytes (*Bryant et al., 1997*; *Bu et al., 2009*; *Cordeiro et al., 2004*; *Domian et al., 2009*), neurons (*Sandoe and Eggan, 2013*) and various types of tumors (*Meacham and Morrison, 2013*). These considerations raise the question of how cellular heterogeneity emerges during embryonic development and contributes to disease formation. Although a number of single-cell transcriptional profiling studies have examined cellular heterogeneity in various cell types, the functional significance of transcriptomic diversity remains challenging to assess. Understanding the cellular processes underlying lineage commitment, lineage diversification, and functional maturation requires the concurrent assessment of transcriptional and functional changes at single-cell resolution. Our imaging technology could therefore complement existing single-cell assays to further our understanding of how transcriptional variation between single cells translates into functional differences.

An unexpected element of heterogeneity we discovered in our study is mRNA localization. Previous studies have demonstrated that in multiple transformed cell lines the β-actin mRNA is largely localized to the cytoplasm with little nuclear presence (*Ben-Ari et al., 2010*; *Buxbaum et al., 2014*; *Femino et al., 1998*; *Nelles et al., 2016*; *Santangelo et al., 2009*). At the same time, a number of studies have suggested that nuclear translocation of the β-actin mRNA might play an important role for normal cell function (*Fallini et al., 2016*; *Smith et al., 2015*).

Concomitant application of MAGIC with the established smFISH technology confirmed the cytoplasmic nature of the β-actin mRNA in U2OS cells; however, strikingly, we found that a fraction of hPSC-derived cardiac myocytes retained the β-actin mRNA in the nucleus (*Figure 2A and C*). What physiologic role could underlie the nuclear retention of mRNA in cardiac myocytes? Gene expression has previously been shown to be subject to stochastic fluctuations arising from short but intense bursts of transcription (*Dar et al., 2012*; *Raj et al., 2006*). Since these bursts can result in profound gene expression noise and thus variations in proteins levels, compartmentalization of mRNA may provide an option to reduce fluctuations in cytoplasmic mRNA levels and may thus be considered a mode of gene regulation (*Bahar Halpern et al., 2015*). Given the imperative role of actin filaments in cardiac myocyte contractility, it is possible that the nuclear retention of the β-actin mRNA ensures stable β-actin protein levels for the correct alignment of actin filaments during the contractility-dependent organization of actin arrays (*Skwarek-Maruszewska et al., 2009*). This possibility is further supported by the interdependence between cellular maturation and contractility (*Ribeiro et al., 2015*) and given that early PSC-derived cardiac myocytes still undergo the process of cellular maturation (*Yang et al., 2014*), it is possible that the cells displaying marked nuclear retention of the β-actin mRNA were in the process of organizing their sarcomeric actin structures. Given the strength

of our technology to provide a spatial resolution of mRNA expression, we anticipate that MAGIC will be useful for the study of the functional significance of variations in mRNA localization.

Lineage assignment in real-time also lends itself to cellular models of biology and disease. Poorly defined heterogeneous mixtures of cell cultures limit the strength of population studies in PSC disease models of specific cellular subtypes. While a number of protocols for efficient PSC differentiation toward specific cell types have been established (*Maroof et al., 2013*; *Pagliuca et al., 2014*; *Protze et al., 2017*), these enrichment strategies are not well suited to study the dynamics governing cellular sublineages. Hence, there is an inherent disconnect between how tissue development and pathogenesis occur in vivo and how they are recreated in PSC in vitro models. Modeling of the biology and disease of specific cell subtypes could now be facilitated using MAGIC by targeting specific cellular subsets using validated markers. Further, lineage assignment in real-time using MAGIC could also potentially be leveraged to select multiple specific cell subtypes by fluorescence activated cell sorting, as previously demonstrated using molecular beacon probes (*Ban et al., 2013*; *Jha et al., 2015*). Our technology preserves cell viability, and mRNA and protein expression after cellular delivery of MAGIC components, which altogether are prerequisites for successful longitudinal studies. Further studies are required to demonstrate the utility of MAGIC for cell sorting purposes.

While our here presented system enables the real-time detection of transcripts, some limitations still remain. Since our technology relies on the delivery of both MAGIC Factor and Probes into the same cell, variable intercellular transfection efficiency may be a limiting factor. This may further be compounded by the cell type of interest, as different cell types may be transfected with variable transfection efficiencies. Given the large number of commercially available transfection reagents, we do not anticipate that efficient transfection of MAGIC constructs poses a considerable challenge to the general applicability of our technology. Additionally, by careful selection of cells based on relative, homogeneous signal intensities, as we did in our study, these limitations could be further alleviated. Collectively, we anticipate that this technology will be useful for any application where cellular heterogeneity with respect to gene expression and cell physiology plays an important biological role.

# Materials and methods

**Key resources table**

| Reagent type (species) or resource | Designation | Source or reference | Identifiers | Additional information |
|---|---|---|---|---|
| Cell line (*H. sapiens*) | H7 | Harvard University | RRID:CVCL_9772 | |
| Cell line (*H. sapiens*) | HUES-9 | Harvard University | RRID:CVCL_0057 | |
| Cell line (*H. sapiens*) | NKX2-5$^{eGFP/w}$ | Dr. Stanley, Monash University, Australia | | |
| Cell line (*H. sapiens*) | U2OS | This study | RRID:CVCL_0042 | Cell line can be obtained from our lab upon request |
| Antibody | Anti-MHCalpha (mouse monoclonal) | R and D Systems | clone 940344 | 1:200 |
| Antibody | Anti-MLC2a (mouse monoclonal) | Synaptic Systems | clone 56F5 | 1:200 |
| Antibody | Anti-MLC2v (rabbit monoclonal) | Proteintech | Cat#: 10906–1-AP | 1:200 |
| Antibody | Anti-Troponin T (mouse monoclonal) | R and D Systems | clone 200805 | 1:1000 |
| Antibody | Anti-Nkx2.5 (goat monoclonal) | R and D Systems | Cat#: AF2444 | 1:400 |

## Molecular cloning, recombinant expression and purification of MAGIC Factor

MAGIC Factor was produced by first amplifying the dsRBD coding site of human protein kinase R (PKR) by PCR and then cloned into pET-14b recombinant expression vector (Novagen) at the NdeI and BamHI restriction sites. The sequence was verified by standard sequencing methods. The pET-14b vector adds a polyhistidine-tag to the N-terminus of the protein to facilitate purification by affinity purification. The polyhistidine-tag was previously shown to not affect the binding affinity of the dsRBD to dsRNA (*Bevilacqua and Cech, 1996*). Recombinant expression was performed by inoculating one clone into LB media containing 20 mM phosphate buffer (pH 7.6), 1% glucose and 200 µg/ml ampicillin and shaking at 37°C for 12 hr. The cells were then pelleted and resuspended in the above media without glucose. The cells were shaken at 37°C until $OD_{600}$ reached 0.3 and then continued shaking at 22°C until $OD_{600}$ reached 0.6–0.8. Protein expression was induced by adding 1 mM IPTG. The flask was shaken at 22°C for another 8 hr before cells were pelleted at 4°C. The cell pellet was frozen overnight and all proteins extracted using 2x CellLytic B reagent (Sigma Aldrich) adjusted to pH 7.2, 0.2 mg/ml Lysozyme, Nuclease (Thermo Scientific) and 2x EDTA-free HALT protease inhibitor cocktail (Thermo Scientific) for 30 min at 4°C. MAGIC Factor was then purified with cobalt immobilized agarose resin (HisPur Cobalt Purification Kit, Thermo Scientific). Binding and wash steps were performed in the presence of 10 mM imidazole, while the bound protein was eluted using 150 mM imidazole. The purified protein was then concentrated into 1.5x PBS (pH 7.4), 1 mM TCEP and 10% Glycerol using Amicon Ultra centrifugal filter units (EMD Millipore) and stored at −80°C in aliquots. Generally, protein concentration and degree of labeling (see below) were determined using a NanoDrop device (Thermo Scientific). The extinction coefficient of the purified protein was calculated to be 12045 $M^{-1}$ $cm^{-1}$ using the online ExPASy tool of the Swiss Institute of Bioinformatics.

## Fluorescent labeling of MAGIC Factor

MAGIC Factor was fluorescently labeled using Alexa Fluor 488 5-SDP Ester (Life Technologies) in 0.1M sodium bicarbonate, pH 8.3 (Sigma-Aldrich) overnight at 4°C. Labeled protein was purified twice into 25 mM HEPES pH 7.2, 150 mM NaCl, 1 mM TCEP, 1 mM EDTA, 10% Glycerol using Zeba Spin Desalting Columns (Thermo Scientific) and concentrated using Amicon Ultra centrifugal filter units (EMD Millipore). Generally, MAGIC Factor was labeled at a 3:1 dye:protein ratio, except for experiments where a range of dye:protein ratio was investigated. For affinity purification of labeled MAGIC Factor (see below), the protein was purified twice into EMSA buffer (see below). Labeled MAGIC factor was stored at 4°C for short-term storage and in aliquots at −80°C for long-term storage.

## Affinity purification of labeled MAGIC Factor

Affinity purification was performed with modifications of a previously published protocol (*Allerson et al., 2003*). Two complementary ssRNA were synthesized (GE Healthcare Dharmacon), of which one carried a –$NH_2$ modification at its 5' end, and annealed at equimolar concentrations in $HEN_{100}$ buffer (25 mM HEPES pH 7.2, 1 mM EDTA, 100 mM NaCl) by heating up to 95°C for 5 min and cooling down to 22°C over 25 min. The dsRNA was then modified with the mid-length cross-linker for amine-to-sulfhydryl conjugation Sulfo-SIAB (Thermo Scientific). Sulfo-SIAB was added at final 50 mM to the dsRNA and the reaction performed in 0.25M $NaHCO_3$ buffer, pH 8.5 for 2 hr in the dark at room temperature. Excess Sulfo-SIAB was removed by standard ethanol precipitation of dsRNA. In parallel, NHS-activated agarose beads (Thermo Scientific) were prepared. All steps were performed at room temperature until the addition of MAGIC Factor, after which all further steps were performed at 4°C. First, the resin was modified with 100 mM cystamine pH 8.0 for 1 hr to turn the amine-reactive group on the beads into thiol-reactive sites. Free amine-reactive sites were blocked with 200 mM Tris, pH 7.5 for 30 min. Cystamine molecules were then reduced with 50 mM TCEP and the modified dsRNA coupled to the beads in sodium phosphate buffer, pH 8.0, 5 mM NaCl overnight at room temperature. Unreacted sites were then blocked with 20 mM iodoacetamide and MAGIC Factor bound to dsRNA in EMSA buffer (see below). Unbound protein was washed away with EMSA buffer and bound protein eluted either with increasing concentrations of KCl or

multiple washes with 1M KCl (for large stock preparation). No dsRNA-controls were prepared by omitting dsRNA at the coupling step.

## Protein polyacrylamide gel electrophoresis (PAGE)

PAGE was performed using standard methods. In general, a 12% resolving and 5% stacking gel was used. Gels were analyzed either by visualizing the native fluorescence of fluorescently-labeled MAGIC Factor or staining the gel with Sypro Orange Protein Gel Stain (Thermo Scientific). Gels were imaged using Typhoon Trio+ imaging system (Amersham Biosciences).

## Generation and purification of MAGIC probes

Sequences of MAGIC Probes were determined based on open mRNA sites using mFold Web Server (http://unafold.rna.albany.edu/?q=mfold/rna-folding-form). MAGIC Probes were generated via in vitro transcription using T7 phage polymerase. For each probe, two complementary single-stranded DNA oligonucleotides containing the promoter sequence for the T7 phage polymerase and the sequence for the probe were designed and commercially purchased (Life Technologies). Both DNA oligonucleotides were annealed in $TEN_{100}$ buffer (10 mM Tris-HCl pH 8.0, 1 mM EDTA, 100 mM NaCl) by heating up to 95°C for 5 min and cooling down to 22°C over 25 min. The dsDNA was used as the template for the in vitro transcription reaction using Megashortscript Kit (Life Technologies). UTP was replaced with aminoallyl-UTP (Thermo Scientific) at the same concentration. 20U SUPERase In RNase Inhibitor (Thermo Scientific) was added to each reaction. In vitro transcription was carried out for 4 hr at 37°C. The reaction was then stopped by adding 1 vol of denaturing gel loading buffer and heated up to 70°C for 10 min. The reaction was then loaded on a 20%/7M Urea denaturing polyacrylamide gel pre-run using 0.5x TBE buffer at 50–55°C. Transcribed RNA was located by UV shadowing. RNA was cut out using sterile scalpels and extracted from the gel overnight at 4°C into TE buffer (10 mM Tris HCl pH 8.0, 1 mM EDTA) containing 0.1% SDS. The next morning, gel slices were pelleted by centrifugation at maximum speed and the supernatant containing the probes transferred into a new tube. Probes were sequentially extracted using Butanol, Phenol/Chloroform/Isoamylalcohol and Chloroform/Isoamylalcohol and precipitated at −20°C for 8 hr using 1/10 vol of 3M sodium acetate, pH 5.2 and 3 volumes absolute ethanol. Probes were pelleted by centrifugation at maximum speed for 30 min at 4°C and washed once with ice-cold 70% ethanol and resuspended in TE buffer (probes that were not further fluorescently labeled) or nuclease-free $H_2O$ (probes that were further fluorescently labeled). RNA was stored at −80°C. In general, probe concentration and degree of labeling (see below) were determined using a NanoDrop device (Thermo Scientific).

## Fluorescent labeling and purification of MAGIC probes

Probes were fluorescently labeled with Alexa Fluor 488 5-SDP Ester, Alexa Fluor 546, 594 or 647 carboxylic acid, succinimidyl ester (Life Technologies) in 0.3M sodium bicarbonate, pH 8.5 (Sigma-Aldrich) for 2 hr at room temperature. Labeled probes were first separated from unreacted dye using Oligo Clean and Concentrator Kit (Zymo Research) and eluted into 10–15 µl TE buffer. Probes were then purified from a 24%/7M Urea denaturing polyacrylamide gel as described above and stored in TE buffer at −80°C.

## Electrophoretic mobility shift assay (EMSA)

EMSA ssRNA probes were generated as described above and the top strand further fluorescently labeled and purified as described above. dsRNA was generated by annealing both ssRNA probes in $TEN_{100}$ and heating the mixture to 95°C for 5 min and then gradually cooling to room temperature over 25 min. Typically, 100–200 nM of RNA was reacted with MAGIC Factor in EMSA buffer (25 mM HEPES pH 7.4, 100 mM KCl, 10 mM NaCl, 0.5 mM EDTA, 1 mM TCEP, 0.1% Nonidet P-40, 5% Glycerol), 10U Superase In RNase Inhibitor (Thermo Scientific) and 0.1 mg/ml t-RNA (Sigma Aldrich) for 30 min at 4°C before loading onto a 12% native polyacrylamide gel containing 2.5% glycerol. The gel was run in 0.5x TBE at 4°C after a pre-run for 60 min at 100V. After sample loading, the gel was run at 100V for 20 min to ensure gentle entry of probes and protein into the gel. The gel was then run at 150V until bromophenol blue from empty lanes reached the front. Gels were analyzed using Typhoon Trio+ imaging system (Amersham Biosciences) and three filter settings. To visualize MAGIC Factor, a blue laser at 488 nm was combined with a 526 nm short-pass filter. To visualize RNA, a red

laser at 633 nm was combined with a 670/30 nm band-pass filter. To visualize FRET, a blue laser at 488 nm was combined with a 670/30 nm band-pass filter. From the images, the relative shift, FRET and FRET/shift ratio were calculated using ImageJ software.

Dissociation constants $K_d$ were calculated from EMSA gels. Unpurified and affinity purified MAGIC Factor were reacted with 200 nM dsRNA at increasing molar ratios and the fraction of dsRNA bound (θ) fitted by nonlinear least squares into

$$\theta = \varepsilon \frac{[\mathrm{MF}]^n}{[\mathrm{MF}]^n + \mathrm{K}_d^n}$$

where ε is the observed maximum fraction bound, [MF] the concentration of MAGIC Factor, $K_d$ the dissociation constant and n the Hill coefficient.

## Characterization of RNA probes with variable degree of labeling

A 20-mer RNA probe with four labeling sites was in transcribed and gel purified as described above. The purified probe was then fluorescently labeled with Alexa Fluor 647 and single- to four-labeled probes gel purified as described above. The fluorescence intensities of the four probes were quantified using a spectrophotometer. Hybridization to unlabeled, complementary RNA was assessed in physiologic buffer resembling cytosolic ion concentrations (25 mM HEPES pH 7.4, 140 mM KCl, 10 mM NaCl, 1 mM $MgCl_2$, 1 mM TCEP) by incubating both strands in equimolar concentrations at 37° C for 30 min or 2 hr. As a positive control, both strands were hybridized at 95°C for 5 min in $TEN_{100}$ buffer as described above for dsRNA. The fractions of hybridized probes were quantified from gels using ImageJ software.

## Cell culture

Human Embryonic Stem Cells (hESC; HUES9, H7 and $NKX2\text{-}5^{eGFP/w}$ lines) were cultured on Matrigel-coated tissue culture polystyrene plates and maintained in Essential eight medium (Thermo Fisher). hESC medium was refreshed every 24 hr and hESCs were passaged at 70% confluency using 0.5 mM EDTA. U2OS cells were cultured in standard DMEM containing 10% FBS and 1% penicillin/streptomycin, and dissociated using TrypLE Express. HUES-9, H7 (Harvard Stem Cell Institue) and Nkx2.5 (eGFP/w) (Dr. Stanly, Monash University Australia) human pluripotent stem cell lines as well as U2OS cells have regularly been tested for the absence of mycoplasma contamination. Their identity have previously been authenticated.

## Cardiac differentiation of hPSC

Cardiac differentiation of hPSCs was induced using small molecules as previously described (*Atmanli et al., 2014*). Briefly, when hPSCs achieved confluency, cells were treated with CHIR99021 (Stemgent) in RPMI (Thermo Fisher) supplemented with Gem21 NeuroPlex without insulin (Gemini Bio Products) for 24 hr (from day 0 to day 1). The medium was replaced with RPMI/G21-insulin at day 1. The cells were then treated with IWP4 (Stemgent) in RPMI/G21-insulin at day 3 and the medium was refreshed on day 5 with RPMI/G21-insulin. Cells were maintained in RPMI supplemented with Gem21 NeuroPlex (Gemini Bio Products) starting from day 7, with the medium changed every 3 days. Ascorbic acid at 50 µg/ml was added to media until onset of beating. Beating clusters were seen starting day 6 of differentiation. Metabolic selection of cardiac myocytes was performed for 3 days by incubating cells in media without glucose but supplemented with 5 mM sodium DL-lactate.

Cardiac myocytes were harvested by treating the cells with Collagenase A and B (Roche) for 5 min first and then TrypLe Express (Thermo Fisher) for another 5 min. Cells were plated onto Matrigel-coated 96-well plates with a No. 1.5 glass bottom or polydimethylsiloxane-coated glass dishes.

## Cellular delivery of MAGIC Factor and probes

MAGIC Factor and Probes were sequentially delivered into cells. Cells were plated at 50,000 cells/$cm^2$ and transfected at 50–60% confluency. At 0 hr, 500 ng MAGIC Factor was mixed with 0.4 µl Pepmute Transfection Reagent (SignaGen) in a total of 10 µl 1x Pepmute Transfection buffer and added to cells in Gem21 media (see above) after a 20 min incubation. At 1 hr, MAGIC Probes were

mixed to 0.75 μM with 0.075 μl TransIT-X2 (Mirus Bio) in a total of 10 μl 1x Pepmute Transfection buffer and added to cells in Gem21 media (see above). Cells were washed and imaged at 2 hr.

### RNA-fluorescence in situ hybridization (RNA-FISH)

Cells were fixed with 4% paraformaldehyde for 10 min and permeabilized with 70% ethanol for at least 2 hr at 4°C. After washing with wash buffer (2xSSC, 10% formamide), β-actin, MHCα or MLC2a MAGIC Probes were added at 37 nM and Stellaris β-actin, MHCα or MLC2a single-molecule FISH (smFISH) probes (Biosearch Technologies) were added at 125 nM to cells in hybridization buffer (2xSSC, 10% dextran sulfate, 1 μg/μl t-RNA, 0.02% BSA, 10% formamide, 40U Superase In RNase inhibitor) for 4 hr at 37°C. Cells were then washed with wash buffer and equilibrated with EMSA buffer. 50 ng purified MAGIC Factor was added to cells in EMSA buffer and reacted for 1 hr at 4°C. Cells were then washed with EMSA buffer and imaged. In RNA-FISH experiments, MAGIC Factor was labeled with Alexa Fluor 488, MAGIC Probe with Alexa Fluor 546 and smFISH probe with Quasar 670 dye.

For quantification of transcript levels, 3D images of cells stained with the β-actin, MHCα or MLC2a smFISH probes were acquired and transcripts levels quantified using Volocity software (Perkin Elmer) as previously described (*Lifland et al., 2011*; *Santangelo et al., 2009*).

### Laser scanning confocal Microscopy

Live-cell imaging was performed on Nikon A1R+ or Leica TCS SP8 confocal systems with live-cell chambers to maintain incubator conditions during image acquisition. For imaging of one FRET pair, the appropriate filters for the donor and acceptors were chosen for sensitized emission technique. MAGIC Factor was excited with a 488 nm laser, and MAGIC Probes were excited with 552 nm and 638 nm lasers, respectively. Single-labeled control samples were used to calculate the relative spectral bleed-through from donor and acceptor fluorophores. Control and experimental samples were imaged using the same imaging settings. For imaging of multiple FRET pairs, spectral imaging and linear unmixing using the built-in module of the microscope software were performed. The entire spectrum was first divided into multiple detection windows. Images in these windows were then acquired by illuminating a field of view subsequently with 488 nm, 552 nm and 638 nm lasers. From single-labeled controls, spectral fingerprints from corresponding fluorophores were obtained during each experiment. Images from samples containing FRET pairs were processed through linear unmixing using these fingerprints to obtain donor, acceptor and FRET images. Since linear unmixing cannot remove acceptor bleed-through signal from FRET images, they were removed manually using ImageJ or Volocity (see below). Control and experimental samples were imaged using the same imaging settings. Single living cells were imaged using a 20x CFI plan-apochromat objective with a numerical aperture of 0.75 and an additional digital zoom. Only cells that showed a homogeneous distribution of fluorescence signal within the cytoplasm were imaged.

### FRET imaging and processing

FRET image processing was performed based on established methods (*Broussard et al., 2013*; *Spiering et al., 2013*). Raw FRET images were background corrected and intensity thresholded within the confocal microscope software to exclude saturated pixels from further analysis, and then aligned to correct for any shifts during acquisition. Images were exported as 12-bit tiff files and further analysis was performed using ImageJ or Volocity software. Images were then masked and a median filter applied to reduce noise during generation of corrected FRET images. The spectral bleed-through constants, calculated from single-labeled control samples, were used to quantify the corrected FRET (cFRET) intensity in co-transfected cells and to generate cFRET images. cFRET is defined as $cFRET = FRET_{raw} - (a \times donor) - (b \times acceptor)$, whereby a and b are the relative spectral bleed-through constants of the donor and acceptor, respectively (*Xia and Liu, 2001*). A Summary of quantified data is provided in *Table 3—source data 1*. All fluorescence intensity quantifications of quantified confocal microscopy data for each channel (MAGIC Factor, MAGIC Probe, FRET) are provided in *Table 3—source data 2*. When spectral imaging was used, only the acceptor bleed-through was further removed from FRET images. For visualization of FRET, a rainbow pseudocolor LUT was applied.

## Calcium imaging

$Ca^{2+}$ imaging was performed following mRNA visualization using MAGIC. Cells were loaded with the fluorescent calcium indicator Fluo-4 AM at 5 µM and spontaneous $Ca^{2+}$ transients of the same cells imaged for gene expression analysis acquired and analyzed using ImageJ software.

## Immunofluorescence

Cells were fixed with 4% paraformaldehyde for 10 min, permeabilized with 0.1% Tween-20 for 20 min and blocked with 10% donkey serum (Sigma-Aldrich) for 1 hr. Primary and secondary antibodies were added to cells in blocking buffer for 2 hr and 1 hr, respectively. Nuclei were counterstained using DAPI. Antibodies used in this article are: myosin heavy chain $\alpha$ (clone 940344, R and D Systems, 1:200 dilution), myosin light chain 2a (clone 56F5, Synaptic Systems, 1:200 dilution), myosin light chain 2 v (rabbit polyclonal, Proteintech, 1:200 dilution), troponin T (clone 200805, R and D Systems, 1:1000 dilution), NKX2-5 (goat polyclonal, R and D Systems, 1:400 dilution), Alexa Fluor-conjugated isotype-specific secondary antibodies (Thermo Fisher, 1:600 dilution).

## Cell viability assay

Cell viability was assessed using Live/Dead cell viability assay kit (Thermo Fisher). For cell viability after single transfection, transfected cells were treated with the assay kit after imaging. For cell viability after co-transfection, transfected cells were assessed after 6 hr and after an additional 18 hr (total 24 hr) of cell culture. Cell counting was performed at randomly selected regions of the culture plate.

## Quantitative PCR

Cells were prepared as for the cell viability assay and RNA extracted at the same timepoints using RNeasy Mini Kit (Qiagen). cDNA was synthesized using iScript cDNA Synthesis Kit (Bio-Rad). Transcript levels were measured and calculated relative to the expression of GAPDH.

## Western blot

Cells were prepared as for the cell viability assay and total protein isolated using Tissue Protein Extraction Reagent (Pierce) at the same timepoints. Proteins were separated and transferred to a Nitrocellulose membrane using Mini-Protean Tetra and Mini Trans-Blot Cell systems (Bio-Rad), respectively. Membranes were blocked in 5% non-fat dry milk, incubated with primary antibodies against $\beta$-actin or $\alpha$-tubulin (Abcam) and probed with horseradish peroxidase-conjugated secondary antibodies (Jackson ImmunoResearch). Membranes were developed with SuperSignal West Pico Chemiluminescent Substrate (Pierce) and analyzed using ImageJ software.

## Statistical analysis

Statistical analysis was performed using Student's *t*-test for differences between two experimental groups and a one-way ANOVA followed by a Tukey post hoc analysis for differences between three or more experimental groups. Differences were considered significant if the p-value was lower than 0.05 ($p < 0.05$).

## Acknowledgements

AA was supported by an American Heart Association Predoctoral Fellowship award (15PRE22220009) and NHLBI Progenitor Cell Biology Consortium (PCBC) Jump Start awards (5U01HL099997-07; Subaward Nos. 10015214 and 101330A). IJD was supported by the NIH/National Heart, Lung, and Blood Institute (U01HL100408-01). AA and IJD are inventors of the pending patent application of the technology described in this manuscript (PCT/US2016/029972).

## Additional information

### Competing interests
Ayhan Atmanli, Ibrahim John Domian: Inventor of a pending patent (PCT/US2016/029972). The other authors declare that no competing interests exist.

### Funding

| Funder | Grant reference number | Author |
| --- | --- | --- |
| American Heart Association | Predoctoral Fellowship: 15PRE22220009 | Ayhan Atmanli |
| National Heart, Lung, and Blood Institute | Progenitor Cell Biology Consortium (PCBC) Jump Start Award: 5U01HL099997-07 | Ayhan Atmanli |
| National Heart, Lung, and Blood Institute | U01HL100408-01 | Ibrahim Domian |

The funders had no role in study design, data collection and interpretation, or the decision to submit the work for publication.

### Author contributions
Ayhan Atmanli, Conceptualization, Data curation, Formal analysis, Funding acquisition, Validation, Investigation, Visualization, Methodology, Writing—original draft, Project administration, Writing—review and editing; Dongjian Hu, Frederik Ernst Deiman, Annebel Marjolein van de Vrugt, Data curation, Formal analysis, Validation, Investigation; François Cherbonneau, Validation; Lauren Deems Black III, Project administration; Ibrahim John Domian, Conceptualization, Funding acquisition, Writing—original draft, Project administration, Writing—review and editing

### Author ORCIDs
Ayhan Atmanli (iD) https://orcid.org/0000-0001-6951-8893

### Decision letter and Author response
Decision letter https://doi.org/10.7554/eLife.49599.023
Author response https://doi.org/10.7554/eLife.49599.024

## Additional files

### Supplementary files
• Transparent reporting form DOI: https://doi.org/10.7554/eLife.49599.021

### Data availability
All data generated or analysed during this study are included in the manuscript and supporting files.

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
