## [Decision Letter]

Thank you for submitting your article "Live single-cell transcriptional analysis demarcates cellular functional heterogeneity" for consideration by *eLife*. Your article has been reviewed by three peer reviewers, including Sean Wu as the Reviewing Editor and Reviewer #1, and the evaluation has been overseen by Didier Stainier as the Senior Editor.

The reviewers have discussed the reviews with one another and the Reviewing Editor has drafted this decision to help you prepare a revised submission.

Summary:

This article describes a new technology for identifying and labeling live single cells in vitro using fluorescent reporter bound to mRNA. The strategy takes advantage of the specificity of fluorescent RNA probe for binding to mRNA transcript of interest and the illumination of bound dsRNA using the dsRNA binding domain of protein kinase R coupled to a fluorescent FRET donor molecule. By transfection into cultured cells, the authors were able to detect fluorescent signal output from single human iPSC-derived cardiomyocytes expressing either MLC2a or MHCα or both. This technology addresses the lack of appropriate tools to identify live single cells by fluorescent imaging and quantification of mRNA expression in living cells.

Overall, the reviewers felt that the experiments were well organized and professionally conducted and the findings are relatively clear and may provide new insights in identifying the cellular phenotypic diversification and understanding the complex biological processes that underlie development and disease. The results from using such tools would be potentially of significant interest to the scientific community.

Essential revisions:

1) An advantage of this method is the ability to multiplex different probes with different fluorescence signals and use this to measure gene expression for multiple targeted genes in the same cell. A demonstration of detecting 3 different transcripts by MAGIC probe in the same cell by RNA-probe FRET would add greater utility to the technology described.

2) Another advantage of this method is to be able maintain cells alive which would enable sorting of fluorescent cells by FACS. Can the authors describe the use of this tool to do cell staining and FACS-based sorting?

3) The demonstration of the progressive increase in detection of sarcomeric genes by MAGIC during iPSC cardiac differentiation is quite nice. Can this technique also detect transcription factors that are lower in abundance?

4) What are the most significant advantages and disadvantages between the existing approaches (Introduction, second paragraph) and the authors' MAGIC approach for live-cell mRNA imaging? Please describe in a table a direct comparisons of this vs. other approaches the main advantages and disadvantages of this new approach.

---

## [Author Response]

Essential revisions:1) An advantage of this method is the ability to multiplex different probes with different fluorescence signals and use this to measure gene expression for multiple targeted genes in the same cell. A demonstration of detecting 3 different transcripts by MAGIC probe in the same cell by RNA-probe FRET would add greater utility to the technology described.

To leverage the power of our technology for the detection of 3 different transcripts, we imaged FRET in single living cells using spectral imaging. We now provide a new paragraph extensively describing this imaging technique as well as proof-of-principle experiments demonstrating the validity of our approach (revised Figure 6A-C, Figure 6—figure supplements 1 and 2). We also provide a proof-of-principle demonstration of the detection of 3 different transcripts in single living cells by our technology. We show how cells differentially express MLC2a, NKX2-5 and MHCα during hPSC differentiation in the revised Figure 6D-F.

2) Another advantage of this method is to be able maintain cells alive which would enable sorting of fluorescent cells by FACS. Can the authors describe the use of this tool to do cell staining and FACS-based sorting?

In our manuscript, we intended to demonstrate the utility of our technology for single-cell analyses to select for individual cell lineages and to assess their physiology in real-time. While MAGIC coupled with FACS would be a powerful tool for the purification of multiple individual cell subtypes, in the current iteration of our technology we did not assess its utility for FACS assays. We do however discuss now the potential of MAGIC for cell sorting assays using FACS in the Discussion. We hope that further studies will demonstrate the utility of MAGIC for FACS.

3) The demonstration of the progressive increase in detection of sarcomeric genes by MAGIC during iPSC cardiac differentiation is quite nice. Can this technique also detect transcription factors that are lower in abundance?

Transcription factors are widely recognized as key effectors of cellular identity and function and are typically expressed at much lower levels than structural proteins (such as MHCα and MLC2a). In the case of the heart, NKX2-5 controls a transcriptional hierarchy that directs cardiac differentiation and is required for normal cardiac function. Accordingly we examined the expression of this low-abundance transcription factor with MAGIC. We show that our technology is able to detect low copy mRNA and can be used to detect the longitudinal expression of transcription factor in developing cells. This set of data is included in Figure 5.

4) What are the most significant advantages and disadvantages between the existing approaches (Introduction, second paragraph) and the authors' MAGIC approach for live-cell mRNA imaging? Please describe in a table a direct comparisons of this vs. other approaches the main advantages and disadvantages of this new approach.

We now provide a table summarizing the key advantages and disadvantages of MAGIC and other live-cell mRNA imaging technologies in the revised Table 1.